# OpenReview forum: "Inference-Time Personalized Safety Control via Paired Difference-in-Means Intervention"
_ICLR.cc/2026/Conference — ICLR 2026 Poster_

### Official Review · Reviewer_gb3V · 2025-10-21

**Soundness:** 2
**Presentation:** 3
**Contribution:** 2
**Rating:** 4
**Confidence:** 2

**Summary:**

This paper proposes a training-free, inference-time activation intervention for personalized safety in LLMs. Building on LLM steering, it instantiates that toolkit specifically for safety control. The authors compare three direction-estimation strategies and provide a theoretical analysis showing that their paired, topic-matched estimator (PCMS) performs best under the stated assumptions. Experiments report stronger harmfulness reduction with competitive utility relative to prompt-based and activation-editing baselines.

**Strengths:**

The method is training-free, it requires no finetuning or weight changes, making it easy to deploy at inference time. The paper provides a comparison of three direction-estimation strategies and a theoretical analysis that largely aligns with the empirical trends. The results show a practical inference-time alternative to post-training approaches like SFT or RLHF.

**Weaknesses:**

1. The method is best viewed as a training-free, LLM steering approach to safety control, paired with PCMS, which leverages topic-matched paired activation contrasts to estimate a robust intervention direction. It is unclear how the proposed approach differs from prior LLM-steering techniques when applied to safety control.
2. While the paper targets personalized safety and advertises a training-free approach, the method still requires model- and facet-specific offline estimation of safety directions from paired prompts, which introduces additional setup cost. In practice, the implementation focuses on four safety facets (e.g., violence, political ideology, sexuality, mental health), so the claimed ‘personalization’ is category-level rather than truly user-specific.
3. The evaluation relies heavily on a single LLM to score both harmfulness and utility. Using multiple independent judges and a task-specific, fine-tuned scoring model (calibrated with human annotations) would be much more robust and credible.
4. The paper does not investigate whether a safety direction estimated on one model transfers to other models. If transfer does not hold, the approach still requires per-model direction estimation and calibration, adding non-trivial setup cost and limiting practical reuse.
5. The paper’s theory broadly tracks the empirical trends but relies on stylized assumptions, most notably an additive decomposition of activations into a topic component plus a harm component with zero-mean noise, together with a linear harm scorer. These choices make PCMS analytically clean, yet the results mainly rationalize the estimator ranking rather than fully explaining why the intervention works in the non-linear, sequence-level setting of real LLMs. The design of PCMS is simple and reasonable, but the theoretical guarantees may not extend beyond these assumptions.

**Questions:**

See Weaknesses.

---

> ### Author Response · Authors · 2025-12-04
> **Response to Reviewer gb3V (Part 1)**
>
> We appreciate the reviewer's recognition of our method's key strengths: its training-free, inference-time deployment, the practical value as an alternative to post-training methods, and the robust theoretical and empirical comparison of estimation strategies.
>
> We address the stated concerns in detail below.
>
> ---
>
> ## W1: Distinction from Prior LLM-Steering Techniques
>
> We thank the reviewer for this opportunity to clarify our positioning. While we build upon the foundational concept of activation steering, our contribution represents a shift from exploratory techniques to a rigorous, deployment-ready system for personalized safety. We will revise the manuscript to explicitly frame these contributions as a cohesive framework, highlighting the following key distinctions:
>
> - **Theoretical foundations for reliable safety:** Unlike exploratory methods that often lack formal guarantees, our framework provides the rigorous assurance necessary for safety-critical domains. We present a formal bias-variance analysis proving that PCMS is an unbiased and consistent estimator of the true harm direction, ensuring the intervention is stable and predictable. Furthermore, we derive a worst-case optimality guarantee for the intervention direction, providing a principled mathematical basis for trusting the model’s behavior that ad-hoc steering methods cannot offer.
>
> - **A comprehensive control system:** We introduce a holistic inference-time control system rather than a standalone steering vector. A raw intervention vector is insufficient for deployment because it lacks context awareness. To address this, we developed a dynamic controller that determines if and how strongly to intervene. By utilizing a quantile-based activation threshold ($T^{(f)}$), our system ensures interventions are only triggered when prompts explicitly approach harmful regions. This transforms steering into a practical safety mechanism that preserves utility for benign queries.
>
> - **Scalability via multi-facet composition:** Real-world personalization requires handling diverse and overlapping user sensitivities simultaneously (e.g., restricting both violence and political content). Our framework extends steering into a compositional system using a weighted combination mechanism. By adaptively weighting safety directions based on per-prompt relevance (Eq. 11), our approach scales to complex, multi-dimensional preference sets without the need for retraining or separate models.
>
> ---
>
> ## W2: Setup Cost and Category-Level vs. User-Specific Personalization
>
> We acknowledge this distinction and clarify our personalization paradigm:
>
> - **Category-level as practical personalization:** Category-level preferences (e.g., 'avoid violence') represent the natural granularity at which users express safety preferences. Truly user-specific directions would require substantial per-user data collection, which contradicts our low-data design goal.
>
> - **Compositional personalization:** Users achieve personalization by selecting and combining from available facets. With $N$ facets, users can express $2^N$ distinct preference profiles. Our framework is extensible—new facets can be added with approximately 150 paired prompts each.
>
> - **Minimal setup cost:** Direction estimation is a one-time offline computation per model-facet pair, requiring only forward passes (no backpropagation). For LLaMA-3.1-8B, estimating all four directions takes approximately 5–7 minutes on a single H100 GPU.
>
> - **Comparison to alternatives:** Methods like fine-tuning or reward model training require orders of magnitude more data and compute. Our approach offers a favorable trade-off between personalization granularity and practical deployment cost.

---

> > ### Author Response · Authors · 2025-12-04
> > **Response to Reviewer gb3V (Part 2)**
> >
> > ## W3: Single LLM Evaluator
> >
> > To ensure that our results are not artifacts of a specific evaluator model, we conducted **cross-model validation using Claude-3.7**. As detailed in Table 6 (Appendix A.8), the results confirmed that PCMS consistently outperforms baselines in the safety-utility trade-off, achieving a substantial reduction in harmfulness while retaining higher utility than competing methods. This confirms that the superior performance of PCMS is a result of the method's effectiveness, not a bias in the evaluator.
> >
> > We also conducted a **human evaluation** on a representative subset covering all four safety categories (Violence, Political Ideologies, Sexuality, and Mental Health). Each response was rated by 5 independent annotators on **Helpfulness (1–10)** and **Harmfulness (1–5)**, comparing Direct Prompting (DP) against PCMS.
> >
> > **Human evaluation results:**
> >
> > | Metric                | DP            | PCMS          |
> > |-----------------------|---------------|----------------|
> > | Helpfulness (1–10)    | 8.25 ± 1.0    | 7.67 ± 1.1     |
> > | Harmfulness (1–5)     | 3.52 ± 0.9    | 1.79 ± 0.7     |
> >
> > These results closely mirror the trends observed in automated evaluations. Human judges confirmed that PCMS significantly reduces harmfulness (3.52 → 1.79) while maintaining high helpfulness. This strong convergence reinforces the validity of our findings and demonstrates that the safety gains of PCMS are perceptible and preferred by human users.
> >
> > We include these additional results in the revision (Appendix A.7).
> >
> > ---
> >
> > ## W4: Cross-Model Transferability
> >
> > We appreciate the reviewer’s interest in cross-model transfer. While this is an interesting direction, we note that:
> >
> > - **Model-specific focus:** Most activation-steering work, including ours, focuses on estimating directions within a single model, which allows precise, model-specific control. Activations are highly model-dependent across architectures or independently trained models, so even conceptually similar directions may not align across different embedding spaces.
> >
> > - **Practical efficiency:** Estimating directions per model is relatively cheap, requiring only a few forward/backward passes on ~150 representative prompts per facet. This one-time, offline computation is lightweight compared to other safety interventions and provides reliable control tailored to each model.
> >
> > Given these points, per-model estimation is both practical and sufficient for real-world deployment, and cross-model transfer, while interesting, is not critical to the main contribution of providing principled, reliable personalized safety.
> >
> > ---
> >
> > ## W5: Theoretical Assumptions
> >
> > We appreciate this nuanced critique and offer the following perspective:
> >
> > - **Role of theory:** Our theoretical analysis is intended to justify the estimator design and explain observed performance trends, rather than fully characterize the internal dynamics of LLMs. Empirical results demonstrate that the theoretical predictions are meaningful in practice.
> >
> > - **Additive model as an approximation:** While LLM activations are inherently non-linear, additive decomposition is a well-established first-order approximation in representation analysis ([1], [2]). The empirical success of linear probing and steering methods further supports the practical utility of this assumption.
> >
> > - **Beyond linear scoring:** The linear harm scorer allows tractable worst-case analysis. Importantly, our interventions demonstrably affect generated text, which is a highly non-linear function, indicating that the estimated directions capture meaningful semantic structure beyond the assumptions of the theory.
> >
> > Overall, we believe that the combination of theory and empirical validation strengthens our contribution, providing principled guidance while remaining practically effective in real-world LLMs.
> >
> > ---
> >
> > References:
> >
> > [1] Mikolov, Tomas, et al. "Efficient estimation of word representations in vector space." arXiv preprint arXiv:1301.3781 (2013)
> > [2] Bolukbasi, Tolga, et al. "Man is to computer programmer as woman is to homemaker? Debiasing word embeddings." Advances in Neural Information Processing Systems 29 (2016)

---

### Official Review · Reviewer_axrk · 2025-10-26

**Soundness:** 3
**Presentation:** 2
**Contribution:** 2
**Rating:** 2
**Confidence:** 3

**Summary:**

This paper proposes an efficient, training-free method for personalized safety control in large language models (LLMs) through inference-time activation intervention, referred to as PCMS. Theoretically and empirically, we demonstrate the superiority of PCMS over existing estimators in steering model outputs toward safety alignment.

**Strengths:**

I find the authors' premise that "safety is subjective" to be well-motivated. Consequently, the research question they pose—How can we equip language models with personalized safety controls that reflect individual sensitivities without retraining, heavy data requirements, or compromising efficiency at inference time?—is both timely and significant.

**Weaknesses:**

+ The description of the methodology lacks clarity in several key aspects. Most notably, it remains unclear how the activations for two of the three proposed strategies are derived or defined, which is critical for understanding and reproducing the approach.
+ The experimental evaluation relies primarily on datasets like SHP and the harmless portion of PHTest, which consist of content with relatively low levels of harm. While I understand the authors' intention to demonstrate personalized safety control, this choice raises concerns about whether the method can ensure safety under more stringent, conventional definitions. Furthermore, the defined safety facets—violence, political ideologies, sexuality, and mental health—largely reflect areas with broad societal consensus. True subjective safety, as illustrated by the "historical revolution" example in the introduction, often arises in more contested contexts such as historical narratives or fictional settings (e.g., games or movies). Evaluating the method on facets where subjectivity is more pronounced would better substantiate the paper's core claim.

**Questions:**

1. If I understand correctly, the activations for two of the strategies are derived from the prompts shown in Table 1. However, the distinction between the "test prompt" and the "reference prompt" is not sufficiently elaborated. A flowchart or a concrete, step-by-step example illustrating how the average harm-difference vector is estimated and applied during inference would greatly clarify this process and enhance the reproducibility of the method.

2. To strengthen the empirical evaluation, I recommend that the authors include benchmark results from established safety datasets. This would more convincingly demonstrate the method's foundational safety alignment capabilities. Specifically, the BeaverTails dataset [1], which offers a comprehensive analysis across fine-grained safety categories, would be a valuable addition. Furthermore, evaluating on the virtual-scenario-based samples from Xstest [2] could effectively showcase the method's ability to perform personalized control in complex, edge-case scenarios. Incorporating these experiments would directly address my primary concerns regarding the method's basic safety guarantees and its efficacy in more challenging, subjective contexts.

3. ​​Minor Points for Clarification:​

   (a) Notation Consistency:​​ Please clarify if $\gamma$ (experiments) and $\alpha$ (Eq. 1) represent the same parameter.

   (b) Figure 3 Axis:​​ The direction of the arrow on the x-axis seems reversed; it might be more intuitive if it pointed left.

[1] Ji, Jiaming, et al. "Beavertails: Towards improved safety alignment of llm via a human-preference dataset." Advances in Neural Information Processing Systems 36 (2023): 24678-24704.

[2] Röttger, Paul, et al. "Xstest: A test suite for identifying exaggerated safety behaviours in large language models." arXiv preprint arXiv:2308.01263 (2023).

---

> ### Author Response · Authors · 2025-12-04
> **Response to Reviewer axrk (Part 1)**
>
> We thank the reviewer for their positive assessment, which strongly validates our core premise that "safety is subjective" and affirms the timeliness and significance of our research question regarding training-free, personalized safety controls.
>
> We address the stated weaknesses and questions in detail below:
>
> ---
>
> ## W1 & Q1: Methodology Clarity
>
> We thank the reviewer for the constructive suggestion. While the theoretical definitions of the estimation strategies and the prompt construction methodology are already detailed in Section 2.2 and Section 3.2 respectively, we include a clear diagram in the revision (Figure 2) to visually delineate the methodology and eliminate any potential ambiguity.
>
> We offer the following clarifications:
>
> 1. **Activation Extraction:** As defined in Section 2.1 (Line 84) and noted in Section 3.3 (Line 414), we extract the hidden state of the last token from the middle layer after a forward pass.
> 2. **Prompt Terminology (Table 1):** The "Test prompt" (yellow) refers to the actual user query evaluated during inference. The "Reference prompts" (blue) are distinct inputs used only in the offline phase to estimate harm directions; they are never seen at inference time.
> 3. **Strategy Implementation (Section 2.2):**
>    - **ILCS:** As defined in Eq. 2, this uses a single difference vector. In the global setting, this comes from a generic reference pair. In the local setting, the single difference vector is derived directly from the test prompt itself by appending different suffixes to create the contrastive pair.
>    - **UMS:** As defined in Eq. 3, this uses the difference between the means of two unpaired collections (clean vs. harmful reference datasets).
>    - **PCMS:** As defined in Eq. 4, this averages the difference vectors of *n* paired reference prompts, where each pair shares a base query but differs in safety suffix.
>
> To further support this, we add a flowchart (Figure 2) in the main text illustrating the separation between the offline estimation phase (using reference prompts) and the online intervention phase of each strategy.
>
> ---

---

> > ### Author Response · Authors · 2025-12-04
> > **Response to Reviewer axrk (Part 2)**
> >
> > ## W2 & Q2: Dataset Choice and Safety Facet Coverage
> >
> > We thank the reviewer for this insightful critique. We agree that personalized safety is most critical in contested, subjective contexts—such as historical narratives or fictional settings—where "safety" is not a universal truth but a matter of individual preference.
> >
> > We address this concern through a two-part argument supported by new experiments on **XSTest** and **BeaverTails**:
> >
> > 1. **Broad facets as control mechanisms for subjective contexts:**
> >    Our "broad" safety facets (e.g., Violence) function as precise control knobs users require to navigate subjective nuances.
> >    - *Example:* In the "historical revolution" scenario, a user employing the "Violence" facet does not intend to erase the historical event but to steer the narrative style away from graphic descriptions. Evaluating these facets on benign, narrative prompts is the most direct way to measure the model's ability to handle subjective safety.
> >
> > 2. **Empirical validation on contested scenarios:**
> >
> >    **2.1 XSTest evaluation:**
> >    We selected 100 prompts from XSTest related to historical events, fictional settings, and video games. We applied the PCMS "Violence" and "Political Ideology" facets to these prompts.
> >
> >    | Category           | Harmfulness (DP) | Harmfulness (PCMS) | Utility (DP) | Utility (PCMS) |
> >    |-------------------|-----------------|------------------|-------------|----------------|
> >    | Violence (History/Fiction) | 3.11            | 1.63 (↓1.48)      | 8.48        | 7.76 (↓0.72)  |
> >    | Political Ideology        | 3.19            | 1.58 (↓1.61)      | 8.52        | 7.69 (↓0.83)  |
> >
> > PCMS effectively steers the model in these complex edge cases, confirming that broad facets successfully regulate subjective narratives. In addition to quantitative results, we include qualitative example in Figure 9 in the revision, demonstrating how activation-level steering can shift the model toward a safer, preference-aligned alternative.
> >
> >    **2.2 BeaverTails evaluation:**
> >    We evaluated PCMS on BeaverTails to demonstrate that it preserves baseline safety while applying personalized constraints.
> >
> >    - **(a) Preservation of refusal (unsafe prompts):**
> >      We evaluated 140 adversarial samples across 14 categories (e.g., hate speech, illegal activities). PCMS maintained the **exact same refusal rate as the baseline (DP)**:
> >
> >      | Metric       | DP    | PCMS  |
> >      |-------------|-------|-------|
> >      | Refusal Rate (%) | 98.57 | 98.57 |
> >
> >    - **(b) Granular steering (safe prompts):**
> >      We evaluated benign prompts that naturally evoke sensitive themes. PCMS successfully reduced undesired content intensity while preserving response utility:
> >
> >      | Category           | Harmfulness - DP | Harmfulness - PCMS | Utility - DP | Utility - PCMS |
> >      |-------------------|-----------------|------------------|-------------|----------------|
> >      | Violence           | 3.24            | 1.76 (↓1.48)      | 8.42        | 7.71 (↓0.71)  |
> >      | Political Ideology | 3.35            | 1.79 (↓1.56)      | 8.56        | 7.65 (↓0.91)  |
> >
> > These results confirm that PCMS achieves **dual behavior**: maintaining strict refusal for adversarial attacks while providing fine-grained, user-aligned steering for benign inquiries.
> >
> > We include these additional experiments in Appendix A.9
> > ---
> >
> > ## Q3: Minor Clarifications
> >
> > - **(a) Notation consistency:**
> >   - **α (Eq. 1):** Intervention magnitude scalar in the single-facet formulation:
> >
> >     $$
> >     X' = X - \alpha \cdot d_{\text{int}}
> >     $$
> >
> >   - **γ (Eq. 8, experiments):** Sensitivity parameter in the multi-facet adaptive mechanism, used to compute dynamic relevance scores $\alpha^{(f)}(X)$ based on the test prompt's proximity to undesired content:
> >
> >     $$
> >     \alpha^{(f)}(X) =
> >     \begin{cases}
> >       \max(0, \gamma - d^{(f)}(X)), & \text{if } d^{(f)}(X) \le T^{(f)} \text{ (within threshold)} \\
> >       0, & \text{if } d^{(f)}(X) > T^{(f)} \text{ (outside threshold)}
> >     \end{cases}
> >     $$
> >
> >     Here, $\gamma$ controls how steeply the intervention strength decays with distance from the undesired content profile.
> >
> >   - **α\_global (Eq. 10):** Set to 1.0 in implementation; the intervention magnitude is fully controlled by the dynamic weights derived from $\gamma$.
> >
> > - **(b) Figure 3 Axis:** This is a typographical error in the figure that will be corrected in the revision.

---

### Official Review · Reviewer_QnDx · 2025-10-28

**Soundness:** 3
**Presentation:** 3
**Contribution:** 2
**Rating:** 4
**Confidence:** 3

**Summary:**

This paper introduces a inference-time activation steering method for personalized safety control by adding vectors that neutralize harmfulness to the model activation. The authors tested three alternative methods to steering: ILCS, UMS, and PCMS, in which PCMS performs the best in both mitigating harm and preserving helpfulness. Compared to prompt-based methods, PCMS also demonstrated overall higher effectiveness: it reduces harm more than DP, but at the cost of helpfulness.

**Strengths:**

The proposed method is novel and targets the important problem of personalized safety. Existing approaches often relies on prompting to steer LLMs for personalization without editing the activations of the model. On the other hand, this paper explores the less used tool of activation engineering to solve this issue. The evaluation shows that the proposed method presents a strong alternative to prompt-based methods, with particularly impressive improvements on harmlessness. The paper is relatively clearly written (though the theoretical analysis section could be made less dense and unpacked more to facilitate understanding).

**Weaknesses:**

Although the proposed method is novel and seems effective, I have a few major reservations about this work:

1. The method is not validated on real human user data, which makes it challenging to judge the usefulness of the approach in real-world practices, since it's framed as a individualized safety control method for users with different values and preferences. At the minimum, whether the cases used in the experiments represent realistic human safety preferences should be validated. Human data could also inform the definitions of the harm facets. I think showing that PCMS works well on real user data will substantially improve the contribution and practical relevance of this work.

2. Missing evaluation on general capability before vs after activation engineering. PCMS still makes the assumption that there exists a universal "harm vector" that is consistent across all topic distributions, which may not be true. The paper only tests 4 categories (violence, political ideologies, sexuality, and mental health), which the steering direction is also computed based on. It's not guaranteed that helpfulness would be equally well preserved in other categories (e.g., general everyday queries). Additional experiments evaluating the general capability of the model before and after activation engineering will provide helpful context.

**Questions:**

1. Figure 3: why does the error next to "Harmfulness Score (Lower is Better" point rightward instead of downward or leftward?

2. Table 3: Although it's clear that PCMS consistently decreases harmfulness from DP across models, it seems that it also consistently impact utility more severely than DP. If you use a composite metric that accounts for both utility and harmlessness (like in figure 3), will PCMS still be consistently better? If so, I suggest using such a metric instead. Alternatively, please explicitly acknowledge the impairment on utility.

3. There is a trade-off between utility and harmlessness - this paper uses a linear combination between the Utility Score and Harmfulness Score (Figure 3) to quantify this tradeoff, arguing that PCMS is better than other methods by this composite metric. Could you justify why this particular, seemingly arbitrary combination of Utility Score (1-10) and Harmfulness Score (1-5) is a good representation of the tradeoff between harmfulness and helpfulness? Again, I think a more robust approach is to gather human preference data on pairs of outputs generated by different methods (e.g., DP vs PCMS) - if annotators prefer PCMS over other methods, this could be strong evidence that PCMS is better than alternatives.

---

> ### Author Response · Authors · 2025-12-04
> **Response to Reviewer QnDx (Part 1)**
>
> We thank the reviewer for their positive comments, especially recognizing the novelty of using activation engineering for personalized safety, its potential as a strong alternative to prompt-based methods, and the impressive improvement in harmlessness.
>
> Our responses to the stated concerns are detailed below:
>
> ---
>
> ## W1: No Validation on Real Human User Data
>
> We appreciate the reviewer’s emphasis on real-world validation. While accessing large-scale, individualized user preference data would be ideal, significant privacy constraints (e.g., GDPR, CCPA) and the sensitive nature of safety alignment currently make such datasets unavailable to the research community.
>
> However, this constraint does not undermine the validity of our approach because our framework targets explicit rather than inferred personalization. We envision a deployment model analogous to "content maturity settings" or "parental controls," where users actively select specific categories to restrict (e.g., "block political content"). Under this model, the critical validation metric is not whether the model can guess a user's latent preferences from their history, but whether the intervention reliably enforces the constraints the user has explicitly selected.
>
> Our experimental design directly validates this usage scenario. The safety facets we selected (Violence, Political Ideologies, Sexuality, Mental Health) are not arbitrary; they align with standard content moderation categories used by major platforms (e.g., Meta’s Community Standards, OpenAI’s Usage Policies), making them highly realistic candidates for a user-facing menu.
>
> To confirm that this approach translates to perceived safety improvements for humans, we conducted a targeted human evaluation study on a representative subset of samples. Annotators were shown the user’s selected constraint and asked to evaluate the model’s compliance. As shown in the results table below, human annotators confirm that PCMS significantly reduces harmfulness while maintaining acceptable helpfulness:
>
> | Metric                | DP            | PCMS          |
> |-----------------------|---------------|----------------|
> | Helpfulness (1–10)    | 8.25 ± 1.0    | 7.67 ± 1.1     |
> | Harmfulness (1–5)     | 3.52 ± 0.9    | 1.79 ± 0.7     |
>
> These results confirm that when a user explicitly selects a safety constraint, PCMS significantly reduces harmfulness (dropping from 3.52 to 1.79) with only a minor impact on helpfulness. This demonstrates that our method is effective for its intended real-world application: providing users with reliable, explicit control over model behavior.
>
> We include these additional results in Appendix A.7 in the revision.
>
> ---
>
> ## W2: Missing General Capability Evaluation
>
> We appreciate this concern, as safety interventions should not come at the cost of general reasoning capabilities. We clarify that our method incorporates two key safeguards against degradation:
>
> **1. Selective Activation:** Our quantile-based threshold (Section 3.3) ensures the intervention is effectively "off" for unrelated queries. For standard reasoning tasks (like math or coding) that do not trigger the safety facets, the intervention vector is applied with near-zero weight, inherently preserving the base model's behavior.
>
> **2. Orthogonality of Facets:** By estimating separate directions for specific facets (e.g., Violence) rather than a global "safety" vector, we minimize interference with unrelated semantic subspaces.
>
> To validate this, we evaluated PCMS on 200 samples from the GSM8K test set using “avoid violence” as the user preference. Across both base models, PCMS showed no statistically significant difference in final answer accuracy compared to the unmodified models. This confirms that PCMS does not degrade reasoning performance on unrelated tasks.
>
> **GSM8K Accuracy:**
>
> | Model                      | No Intervention (%) | PCMS (%) |
> |----------------------------|----------------------|----------|
> | LLaMA-3.1-8B               | 80.23                | 80.12    |
> | DeepSeek-LLaMA3-8B         | 83.53                | 83.59    |
>
> We include these additional results in Appendix A.10 in the revision.

---

> > ### Author Response · Authors · 2025-12-04
> > **Response to Reviewer QnDx (Part 2)**
> >
> > ## Q1: Figure 3 Arrow Direction
> >
> > Thank you for catching this. This is a typographical error in the figure that we will correct in the revision.
> >
> > ---
> >
> > ## Q2 & Q3: Utility Impairment and Trade-offs
> >
> > We explicitly acknowledge in Section A.11 (Limitations) that safety interventions inevitably incur a utility cost, characterizing this as a fundamental and open challenge in alignment. While this trade-off is unavoidable, we contend that PCMS represents the superior approach based on two robust criteria:
> >
> > 1. **Pareto Optimality:** We agree that fixed composite scores are inherently arbitrary, as different stakeholders prioritize safety and utility differently. Therefore, the robust measure is Pareto optimality. As shown in Figure 3, PCMS lies on the efficient frontier, strictly dominating methods like ILCS and offering a far better exchange than UMS (which sacrifices severe utility for marginal safety gains). This indicates that PCMS is an optimal intervention candidate regardless of the specific weights assigned to safety versus utility.
> >
> > 2. **Human Validation:** Following the reviewer's suggestion for a more robust approach, our human evaluation confirms that this trade-off is desirable in practice. Human judges found that the substantial reduction in harmfulness (dropping from 3.52 to 1.79) justified the minor dip in helpfulness, validating that PCMS is preferred for safe deployment.
> >
> > ---

---

### Official Review · Reviewer_LPsB · 2025-10-31

**Soundness:** 2
**Presentation:** 3
**Contribution:** 2
**Rating:** 6
**Confidence:** 2

**Summary:**

This paper studies personalized safety control for LLMs. They steer the LLM outputs toward safer responses by adding to the LLM activations at inference time, making this a training-free method. In particular, their steering direction approximates the personalized harm directions, rather than move along pre-defined behavior axes. They propose three different strategies: ILCS, UMS, and PCMS, argue that the first two have issues while PCMS tries to combine the strengths of those two. They conduct studies on topics in violence, politics, sexuality, and mental health and show that their method is superior.

**Strengths:**

* The problem of safety alignment in LLMs is an increasingly important topic as LLMs become ubiquitous.
* The proposed approach is lightweight since it does not require training and directly steers LLM activations at inference time.
* The experimental evaluation shows competitive performance of PCMS, achieving higher utility and lower harmfulness than baseline methods.

**Weaknesses:**

* I feel a few of the mentioned limitations should actually be ablations for this work instead of future work. For example, the authors choose the middle layer activations to conduct steering, but it would be very helpful to investigate how the performance shifts when steering on different layers.
* In addition, I think the authors should conduct an ablation on different step sizes α and comparing them to the proposed multi-facet safety control.

**Questions:**

* Line 080 and 093: Is $d_{\text{int}}$ overloaded? It's the optimal unit direction and also harm direction.
* Line 170: how is this additive assumption enforced? It seems model-dependent.
* Figure 2, what is the color bar?

---

> ### Author Response · Authors · 2025-12-04
> **Response to Reviewer LPsB**
>
> We thank the reviewer for their positive assessment, which recognizes the importance of the personalized safety problem, the lightweight and training-free nature of our inference-time steering approach, and the competitive performance of PCMS in achieving a superior utility-harmfulness trade-off.
>
> ---
>
> ## Regarding ablations (Weaknesses)
>
> We thank the reviewer for emphasizing the importance of these ablations. We would like to highlight that both requested ablations are already comprehensively included in our submission (Section A.3):
>
> **Layer selection ablation:** We evaluated performance across all transformer layers (4–32). Crucially, the results in Figure 6 demonstrate consistent performance across mid-to-late layers (12–32). This wide operating window means that hyperparameter tuning effort is mild: the method is robust rather than brittle, allowing practitioners to deploy it effectively without needing precise, resource-intensive optimization.
>
> **Intervention magnitude ablation:** As Figure 7 demonstrates, our analysis of γ across the range [0.0, 1.8] reveals a wide effective range (γ ∈ [0.8, 1.2]) that consistently maintains a desirable balance between harmfulness reduction and utility preservation.
>
> ---
>
> ## Regarding Questions
>
> **Q1: Is $d_\text{int}$ overloaded? (Lines 080, 093)**
> $d_\text{int}$ refers to the same concept throughout: it is the unit intervention direction that encodes the harm direction. We use "optimal unit direction" (line 80) to emphasize it is normalized (||$d_\text{int}$|| = 1), and "harm direction" (line 93) to describe its semantic meaning (it points in the direction of harmful content in activation space). These are not two different objects—$d_\text{int}$ is simply the normalized harm-difference vector used for intervention. We will revise the text to consistently refer to $d_\text{int}$ as the normalized harm direction used for intervention, ensuring clarity across the manuscript.
>
> **Q2: How is the additive assumption enforced? (Line 170)**
>
> The additive assumption (Eq. 5) is not an enforced constraint but a modeling assumption adopted for theoretical tractability, consistent with standard practices in representation analysis. Classical concept-direction methods such as TCAV [1] model concepts as linear vectors in activation space, and linear probing approaches ([2], [3]) likewise rely on the assumption that semantic attributes are approximately linearly separable. More recent activation-engineering work—including contrastive activation addition ([4]), representation engineering ([5]), and difference-in-means behavioral directions ([6])—also adopts this linear directional structure. While this assumption abstracts away some nonlinear and entangled structure present in real LLM representations, as evidenced by both prior work and our own empirical results, this linear modeling framework is sufficiently expressive for identifying effective intervention directions in practice.
>
> References:
> [1] Kim, Been, et al. "Interpretability beyond feature attribution: Quantitative testing with concept activation vectors (TCAV)." International conference on machine learning. PMLR, 2018.
> [2] Alain, Guillaume, and Yoshua Bengio. "Understanding intermediate layers using linear classifier probes." arXiv preprint arXiv:1610.01644 (2016).
> [3] Hewitt, John, and Percy Liang. "Designing and interpreting probes with control tasks." arXiv preprint arXiv:1909.03344 (2019).
> [4] Turner, Alexander Matt, et al. "Steering language models with activation engineering." arXiv preprint arXiv:2308.10248 (2023).
> [5] Zou, Andy, et al. "Representation engineering: A top-down approach to AI transparency." arXiv preprint arXiv:2310.01405 (2023).
> [6] Arditi, Andy, et al. "Refusal in language models is mediated by a single direction." Advances in Neural Information Processing Systems 37 (2024): 136037-136083.
>
> **Q3: Figure 2 color bar**
> The color bar indicates the number of samples in each category or category pair. Diagonal entries show single-facet counts (e.g., 347 sexuality samples), while off-diagonal entries show overlapping samples (e.g., 22 samples tagged as both sexuality and violence).

---

### Official Review · Reviewer_tFbc · 2025-11-01

**Soundness:** 3
**Presentation:** 3
**Contribution:** 3
**Rating:** 6
**Confidence:** 2

**Summary:**

This paper introduces a training-free method for personalized safety control in LLMs. The authors argue that universal safety alignment fails to account for individual user sensitivities. To address this, they propose an inference-time activation intervention framework that modifies internal model activations to suppress undesired content based on user-defined preferences (e.g., related to politics, violence). The main contribution is the Paired Contrast mean shift method, which estimates a "harm direction" in activation space using paired examples of harmful and harmless prompts. This direction is then used to steer model outputs away from undesired content during inference

**Strengths:**

-Works at inference time without retraining or large datasets.
-Reduces harmful content while keeping responses useful.
-Provides formal proof that PCMS is unbiased and optimal under mild assumptions.

**Weaknesses:**

-No automated way to pick the best layer for intervention.
-Sometimes makes answers too cautious or less helpful.
- Relies on LLM-as-a-judge scoring, not human judgment, no ablation for the scoring was given.

**Questions:**

- How reliable is the scoring mechanism? Could the authors manually match a subset of the evaluation set to see how reflective the GPT scores are? Or use two different models for this part to enable some “nuance” to reflect the personalization aspect.
- How could layer selection be improved in future work?

---

> ### Author Response · Authors · 2025-12-04
> **Response to Reviewer tFbc**
>
> We thank the reviewer for their positive points, which highlight our method's ability to work at inference time without retraining, its success in reducing harmful content while maintaining useful responses, and the formal proof that PCMS is unbiased and optimal under mild assumptions.
>
> Our responses for the stated concerns are detailed below:
>
> ---
>
> ## W1 & Q2: Regarding layer selection
>
> We thank the reviewer for highlighting this concern, which we explicitly acknowledge in Section A.7 (Limitations). We offer the following clarifications:
>
> **1. Empirical robustness:** As shown in Figure 5, our method performs consistently well across mid-to-late layers (12-32), with relatively minor performance differences. This indicates that intervention at a middle layer (such as Layer 16, which achieved the best trade-off in our experiments) serves as a robust default without requiring extensive tuning.
>
> **2. Automated Selection:** The layer selection pipeline can be easily automated. Rather than requiring manual intervention, the optimal layer can be identified by evaluating performance on a small validation set, ensuring adaptability to new models.
>
> ---
>
> ## W2: Regarding overly cautious responses
>
> We acknowledge that safety interventions inherently involve a trade-off between reducing harm and preserving utility. However, PCMS is specifically designed to minimize unnecessary caution through two key mechanisms:
>
> **1. Selective intervention:** Our threshold-based activation ensures that the intervention only triggers when a prompt's activation falls within the 98th percentile of proximity to harmful content. This prevents 'blanket caution,' allowing the vast majority of benign queries to pass through unmodified.
>
> **2. User calibration:** Users retain control over this trade-off via the intervention strength parameter γ. As shown in Figure 6, decreasing γ progressively restores utility at the cost of allowing marginally more risk, enabling users to calibrate the model to their specific tolerance.
>
> Empirically, this design achieves the most effective balance among all tested methods. As shown in the Pareto plot (Figure 3) and Table 2, PCMS substantially reduces harmfulness to 1.73 (from 3.36 in Direct Prompting) while retaining a high utility of 7.95. This is a significant improvement over baselines like UMS, which achieves lower harmfulness (1.42) but comes at a severe cost to utility (5.80), suggesting that our method’s caution is principled rather than excessive.
>
> ---
>
> ## W3 & Q1: Regarding the scoring mechanism reliability
>
> To ensure that our results are not artifacts of a specific evaluator model, we conducted **cross-model validation using Claude-3.7**. As detailed in Table 6 (Appendix A.8), the results confirmed that PCMS consistently outperforms baselines in the safety-utility trade-off, achieving a substantial reduction in harmfulness while retaining higher utility than competing methods. This confirms that the superior performance of PCMS is a result of the method's effectiveness, not a bias in the evaluator.
>
> We also conducted a **human evaluation** on a representative subset covering all four safety categories (Violence, Political Ideologies, Sexuality, and Mental Health). Each response was rated by 5 independent annotators on **Helpfulness (1–10)** and **Harmfulness (1–5)**, comparing Direct Prompting (DP) against PCMS.
>
> **Human evaluation results:**
>
> | Metric                | DP            | PCMS          |
> |-----------------------|---------------|----------------|
> | Helpfulness (1–10)    | 8.25 ± 1.0    | 7.67 ± 1.1     |
> | Harmfulness (1–5)     | 3.52 ± 0.9    | 1.79 ± 0.7     |
>
> These results closely mirror the trends observed in automated evaluations. Human judges confirmed that PCMS significantly reduces harmfulness (3.52 → 1.79) while maintaining high helpfulness. This strong convergence reinforces the validity of our findings and demonstrates that the safety gains of PCMS are perceptible and preferred by human users.
>
> We include these additional results in the revision (Appendix A.7).

---

### Author Response · Authors · 2025-12-04
**Summary of Revisions and Response to Reviewers**

We thank all reviewers for their thoughtful feedback. The reviews recognize several key strengths of our work:

## Recognized Strengths Across Reviews:

- **Practical deployment value:** All reviewers acknowledged our training-free, inference-time approach as a practical alternative to costly post-training methods (R_tFbc, R_LPsB, R_QnDx, R_gb3V)

- **Theoretical rigor:** Multiple reviewers praised our formal analysis proving PCMS is unbiased and optimal under stated assumptions (R_tFbc, R_QnDx, R_gb3V)

- **Strong empirical results:** Reviewers noted our method effectively reduces harmful content while maintaining utility, with impressive harmfulness reduction (R_tFbc, R_QnDx, R_gb3V)

- **Important problem:** Reviewers validated that personalized safety addresses a timely and significant challenge (R_QnDx, R_axrk, R_gb3V)

---

## Addressing Reviewer Concerns:

We have comprehensively addressed all raised concerns through additional experiments and clarifications:

- **Evaluation robustness (R_tFbc, R_QnDx, R_gb3V):**
  We conducted both cross-model LLM judge validation (Claude-3.7, Appendix A.8) and human evaluation with 5 annotators showing strong agreement with automated metrics (Appendix A.7).
  Results confirm PCMS significantly reduces harmfulness (3.52→1.79) with modest utility cost (8.25→7.67).

- **Methodology clarity (R_axrk):**
  We added a comprehensive flowchart (Figure 2) illustrating the complete pipeline for all three estimation strategies, clearly distinguishing offline direction estimation from online inference intervention.

- **Broader safety validation (R_axrk, R_QnDx):**
  We conducted additional experiments on established benchmarks:
  - **XSTest:** Demonstrates effective content steering in contested contexts (fictional scenarios, historical narratives) where subjectivity matters most—directly addressing the reviewer's core concern about “true subjective safety.”
  - **BeaverTails:** Confirms preservation of baseline refusal mechanisms while enabling fine-grained personalized control.
  - **GSM8K:** Shows no degradation on general reasoning capabilities (Appendix A.10), confirming the effectiveness of our context-aware control mechanism.

- **Ablation studies (R_LPsB):**
  We clarified that comprehensive ablations are already included (Figures 5–6, Section A.3) showing robust performance across layers 12–32 and intervention magnitudes γ ∈ [0.8, 1.2].

- **Positioning and contribution (R_gb3V):**
  We strengthened our framing to emphasize that—rather than merely applying existing steering techniques—we provide systematic components making activation steering viable for safety-critical deployment:
  1. Theoretical foundations with formal guarantees
  2. Adaptive mechanisms determining when to appropriately intervene
  3. Compositional framework for multiple facets

---

## Revision Highlights:

All new content, including additional experiments and discussion clarifications, and improved positioning, is marked in **blue text** throughout the revision for easy identification. These additions strengthen an already solid contribution by providing additional empirical validation across diverse benchmarks and use cases, demonstrating the robustness and practical applicability of our principled approach.

The core contribution—a theoretically grounded, practically deployable framework for personalized safety—remains compelling and addresses a critical gap in current LLM safety research. We believe these revisions comprehensively address reviewer concerns while maintaining the paper's original strengths.

---

### Meta-Review · Area_Chair_yTLb · 2025-12-17

**Summary:**

This paper presents a training-free method for personalized safety control in large language models (LLMs) through inference-time activation intervention. The authors argue that existing safety alignment approaches often overlook individual sensitivities, leading to the proposal of the Paired Contrast Mean Shift (PCMS) method, which effectively estimates safe intervention directions to suppress undesired content based on user-defined preferences. Empirical evaluations demonstrate that PCMS outperforms other strategies in reducing harmful content while maintaining model utility across various sensitive topics, offering a more adaptive approach to LLM behavior.

**Reviewer Concerns:**

Most of the weaknesses identified by reviewers were noted and addressed by the authors. Here’s a summary of the main areas for improvement:

- *Lack of Clarity in Methodology*: Reviewers felt that the methodology was unclear in several key aspects, particularly regarding how activations for two of the proposed strategies are derived or defined. **Authors took several steps to improve the clarity of the methodology description, notably adding a flowchart (Fig 2).**
- *Confounding Evaluation Metrics*: Reviewers pointed out that only testing a few categories (violence, political ideologies, sexuality, mental health) does not guarantee that helpfulness would be preserved across all topic distributions, such as general everyday queries. Additional evaluations on general capabilities were recommended. **Authors conducted a pilot experiment on GSM8K that showed math capabilities are not affected by this steering, demonstrating that this concern might not be significant.**
- *Dependence on LLM-as-a-Judge*: Some reviewers criticized the reliance on LLM-as-a-judge for scoring, arguing that it lacks the robustness of human evaluations. **Authors added a human evaluation of harmfulness and helpfulness of the steered response versus direct prompting, showing improved harmfulness reduction, adding credibility to their results.**
- *Human Evaluation Limitations*: One reviewer noted that the absence of validation on real human user data makes it difficult to assess the practical usefulness of the proposed method. They suggested that showing results on real user data would enhance the contribution and relevance of the work. **Authors argue that doing this with real human data would be challenging, but their added human evaluation does lend credence to their results.**
- *Scope of 'Personalization'*: Reviewers questioned the claimed personalization aspect of the method, highlighting that it relies on category-level adjustments rather than truly user-specific ones. **Authors addressed this by clarifying that personalization refers to users being able to flip knobs for each of the different categories.**
- *Limited Comparison with Prior Techniques*: Reviewers felt that the differentiation from prior LLM-steering techniques was not clearly articulated. **Authors largely addressed this in their rebuttal, which clarifies the distinctions.**
- *Theoretical Assumptions*: Some reviewers noted that the theoretical foundations of the method relied on stylized assumptions that might limit its applicability in real-world scenarios. **Authors somewhat addressed this in their rebuttal.**
- *Need for Transferability Testing*: The adaptability of safety directions estimated on one model to others was not evaluated, raising concerns around practical applicability. **The AC feels that this is for future work to address, and not within the current paper's focus.**

**Reviewer Scores:**

I think all reviewers would have changed their scores.

---

### Decision · Program_Chairs · 2026-01-26

Accept (Poster)